# Novel Anthraquinone Compounds Inhibit Colon Cancer Cell Proliferation via the Reactive Oxygen Species/JNK Pathway

**DOI:** 10.3390/molecules25071672

**Published:** 2020-04-04

**Authors:** Yuying Li, Fang Guo, Yingying Guan, Tinggui Chen, Kaiqing Ma, Liwei Zhang, Zhuanhua Wang, Qiang Su, Liheng Feng, Yaoming Liu, Yuzhi Zhou

**Affiliations:** Key Laboratory for Chemical Biology and Molecular Engineering of Ministry of Education, Institute of Biotechnology, Shanxi University, Taiyuan 030006, China; 201723002004@email.sxu.edu.cn (F.G.); yanning654@126.com (Y.G.); Chentg@sxu.edu.cn (T.C.); makaiqing@sxu.edu.cn (K.M.); lwzhang@sxu.edu.cn (L.Z.); zhwang@sxu.edu.cn (Z.W.); suqchem1984@163.com (Q.S.); lhfeng@sxu.edu.cn (L.F.); liuym1022@sxu.edu.cn (Y.L.); zhouyuzhi@sxu.edu.cn (Y.Z.)

**Keywords:** anthraquinone derivatives, apoptosis, 3D-QSAR, ROS-JNK, HCT116

## Abstract

A series of amide anthraquinone derivatives, an important component of some traditional Chinese medicines, were structurally modified and the resulting antitumor activities were evaluated. The compounds showed potent anti-proliferative activities against eight human cancer cell lines, with no noticeable cytotoxicity towards normal cells. Among the candidate compounds, 1-nitro-2-acyl anthraquinone-leucine (**8a**) showed the greatest inhibition of HCT116 cell activity with an IC_50_ of 17.80 μg/mL. In addition, a correlation model was established in a three-dimensional quantitative structure-activity relationship (3D-QSAR) study using Comparative Molecular Field Analysis (CoMFA) and comparative molecular similarity index analysis (CoMSIA). Moreover, compound **8a** effectively killed tumor cells by reactive oxygen species (ROS)-JNK activation, causing an increase in ROS levels, JNK phosphorylation, and mitochondrial stress. Cytochrome c was then released into cytoplasm, which, in turn activated the cysteine protease pathway and ultimately induced tumor cell apoptosis, suggesting a potential use of this compound for colon cancer treatment.

## 1. Introduction

Cancer is a major cause of death in populations all over the world. Despite being the current treatment of choice in many cases, chemotherapy still has many disadvantages, such as strong toxic side-effects and partial lack of targeting specificity [1,2,3]. Therefore, the design and synthesis of new compounds that act specifically on target proteins in tumor cells is a focus of current research [4,5,6]. Anthraquinone derivatives with a 9,10-anthracene skeleton are a class of natural compounds frequently used in traditional Chinese medicine. These compounds have various biological properties such as antibacterial, analgesic, antimalarial and antitumor activity [7,8]. In recent years, anthraquinone derivatives have attracted increasing attention due to their excellent antitumor activity. Although anthraquinone derivatives exist in a variety of plants, the extraction process is complicated and problematic due to their low content in plants, which limits further structural optimization and activity screening. Structural modifications are carried out on the basis of the parent nucleus to obtain a drug with stronger efficacy and few side-effects. Indeed, some anthraquinone derivatives have been licensed for use as antitumor drugs in the clinic. Currently marketed compounds, such as mitoxantrone, emodin, doxorubicin and daunorubicin (Figure 1A–D), are used clinically to treat a variety of cancers [9,10]. One of these anthraquinone antitumor drugs, mitoxantrone (MX), has been widely used in the clinic since the 1980s, and is the only relapse**-**remitting treatment approved by the U.S. Federal Drug Administration for worsening of multiple sclerosis symptoms [11]. However, clinical studies have shown that MX causes cumulative and irreversible cardiotoxicity, as most of the MX (>95%) binds rapidly to plasma proteins when it enters the bloodstream [12]. Once there, it causes serious toxic side-effects such as myelosuppression and gastrointestinal reactions, which limit its clinical application [13]. Emodin is another anthraquinone obtained from natural plants that has attracted much attention due to its many pharmacological properties [8,14]. However, due to its poor water solubility, its use is limited and further evaluation is required to determine the full spectrum of its pharmacological activity and toxicity. 

The possibility of combining amino acids with various anti**-**tumor drugs to improve their water solubility and tumor selectivity is currently under evaluation. The demand for protein-forming amino acids in tumor cells is much higher than that of normal cells, which makes amino acids excellent vectors for selective targeting of tumors [14]. Hsin et al. designed and synthesized a series of 1,4-bis(2-aminoethylamino)anthraquinone**-**amino acid derivatives and compared their antitumor properties to MX-amino acid derivatives [15]. Methionine anthraquinone derivatives were found to have lower activity compared to MX-amino acids derivatives, while lysine anthraquinone derivatives had the highest activity. Xu et al. then combined ligustrazine**-**birch acid with amino acids and dipeptides, which raised its antitumor activity compared to unmodified ligustrazine**-**betulinic acid [16].

Activation of the mitochondria-mediated endogenous apoptotic pathway decreases the mitochondrial membrane potential, which leads to a significant increase in mitochondrial membrane permeability. Then it causes the release of cytochrome c. Cytochrome c is released into the cytosol and activates a family of caspases. Then a series of cascade reactions are triggered [17,18,19]. Studies have shown that the increased production of reactive oxygen species (ROS) also induces a decrease in the mitochondrial membrane potential, followed by the release of cytochrome c, which ultimately leads to apoptosis [20,21]. In addition, ROS induce activation of the JNK signaling pathway, which is a key mediator of apoptosis [22,23].

In this study, we synthesized a series of novel amide anthraquinone-amino acid derivatives using 2-methylanthraquinone as a raw material. We carried out structural modifications at the C-1 position, and combined several different amino acids at the C-2 position. All newly synthesized compounds were identified by infrared (IR) and nuclear magnetic resonance (NMR) spectroscopy. The most active compounds were screened by MTT assay and their half-maximal inhibitory concentration (IC_50_) values were determined. We then established an optimal model a three**-**dimensional quantitative structure**-**activity relationship (3D-QSAR) to correlate biological activity with the chemical structure of the synthetic series. Compound **8a** showed good antitumor activity, inducing significant apoptosis via JNK activation in HCT116 cancer cells.

## 2. Results

### 2.1. Chemistry

A series of novel anthraquinone**-**amino acid derivatives **8a**–**8h** were synthesized following the general procedures outlined in Scheme 1.

Initially, 2**-**methylanthraquinone was prepared as a starting material. Subsequently, 2**-**methyl anthraquinone was subjected to nitration under acidic conditions to form 1**-**nitro**-**2**-**methyl anthraquinone. This compound was then oxidized to form 1**-**nitro**-**2**-**carboxy anthraquinone by the addition of Na_2_Cr_2_O_7_. This reaction product was further subjected to acylation to form 1**-**nitro**-**2**-**benzoyl anthraquinone. Finally, a series of 1**-**nitro**-**2**-**acyl anthraquinone**-**amino acids **8a**–**8h** were synthesized by reactions with several different amino acids, including leucine, valine, phenylalanine, glycine, proline, alanine, methionine and glutamic, respectively.

### 2.2. Compound ***8a*** Inhibits Tumor Cell Proliferation But Does Not Affect Normal Cells

The effects of the synthesized amide anthraquinone derivatives on the proliferative activity of eight human cancerous cell lines (EC9706, MCF**-**7, SGC-7901, HepG2, QBC939, HeLa, HCT116 and SW480) in addition to two normal human cell lines (colorectal FHC and liver HL7702) were evaluated using the standard MTT assay. The median IC_50_ values for each compound are presented in Table 1.

Some of the compounds **8a**–**8h** significantly inhibited the proliferative capacity of the tested tumor cells, with IC_50_ values ranging from 21.01 to 72.88 μg/mL, all while exerting a minimal effect on normal liver cells (HL7702) and normal colorectal cells (FHC). It is worth noting that compound **8a** showed the greatest inhibition of HCT116 cells proliferation with an IC_50_ of 17.80 μg/mL. In addition, the IC_50_ of compound **8a** for SGC-7901 cells was 55.82 μg/mL, which was higher than that for the other tumor cells, indicating that this cell line was less sensitive to compound **8a** than the other tumor cell lines. The IC_50_ of **8b** on HCT116 was 21.01 μg/mL, indicating that compound **8b** also exerted strong proliferation suppression activity on this cell line. In contrast, QBC939 cells were less sensitive to compound **8b** than the other tumor cells lines, with an IC_50_ of 72.88 μg/mL. SGC**-**7901 cells were also less sensitive to **8c**, but this compound showed higher suppression activity in HeLa cells, with an IC_50_ of 25.94 μg/mL. MCF-7 was more sensitive to compound **8d** than other cells, although this compound was less suppression activity to SGC-7901 cells. Compound **8e** had the weakest effect on SGC-7901 with an IC_50_ of 60.49 μg/mL. Finally, compounds **8f, 8g**, and **8h** all significantly inhibited the proliferation of HCT116 cells. These data indicate the activity of the anthraquinone derivative with leucine as the β-substituent on the anthraquinone ring is superior to that of anthraquinone derivatives with other amino acid. 

Figure 2A shows the parent compound structure. Compared with the structure of the rhein valine [24], the hydroxyl group at position 8 is removed, the hydroxyl group at the position 1 is replaced with a nitro group, and the amino acid group at the meta position is transferred to the ortho position. In addition, there was a significant difference in inhibitory activity of compounds with substitution of the same amino acid at different sites. The rhein valine IC_50_ was 33.51 μg/mL, while the IC_50_ of **8b** was significantly enhanced at 27.87 μg/mL (Figure 2B,C). The nitro group is a strong electron withdrawing group, while the hydroxyl group is an electron donating group.

### 2.3. 3D-QSAR

To better understand the structure-function relationship, we conducted a 3D-QSAR study to a series of amide anthraquinone derivatives using CoMFA and CoMSIA. The pIC_50_ values (−log IC_50_) of HCT116 cells were selected for 3D-QSAR analysis according to the results of the MTT assay of these compounds, which were found to be superior to the other cell types tested (Table 1). The parameters of the CoMFA and CoMSIA models indicated good relevance and predictive ability (CoMFA: *R^2^* = 0.965, *q^2^* = 0.525, two components; CoMSIA: *R^2^* = 0.983, *q^2^* = 0.6, two components); the contribution of the steric and electrostatic fields was 80: 20. The predicted and experimental values obtained for the best CoMFA and CoMSIA models for each compound are shown in Table 2. 

The correlation between the training-set proliferation suppression activity against HCT116 cells is shown in Figure 3. The results indicated good predictability of the molecules under investigation. In summary, the parameters of this CoMFA and CoMSIA models were found to be reliable and to provide a credible guarantee that novel amide anthraquinone compounds with strong anti-tumor activity and few side-effects can be easily designed.

Figure 4 shows a three**-**dimensional equation of the stereo and electrostatic fields of a CoMFA model with **8a** as the template. As shown in Figure 4A, the CoMFA stereograph shows that the position of the terminal carboxyl group at the side-chain of the anthraquinone ring in compound **8a** is at a favorable position. This indicates that the introduction of carboxyl groups at this position significantly increases biological activity. In addition, the CoMFA electrostatic diagram shown in Figure 4B demonstrates that the carboxyl group in compound **8a** is close to the blue region, which is a desirable trait for increasing the antiproliferative activity. The red regions around the nitrogen atoms in **8a** indicate that it also contributes to antiproliferation activity against tumor cells by increasing the negative charge of the compound.

Figure 5 also shows a three-dimensional equation of the stereo and electrostatic fields of a CoMFA model with **8a** as the template. In this CoMFA model (Figure 5A), the C-terminal in the leucine side-chain and on the left of the anthraquinone ring has yellow areas, which indicate that the introduction of small molecular groups in this region can increase the antiproliferative activity. The CoMSIA model electrostatic field equipotential map illustrated in Figure 5B shows the anthraquinone ring and leucine side-chain in compound **8a** surrounded by a blue equipotential domain, which is a desirable trait for increasing antiproliferative activity.

For the hydrophobic CoMSIA model, the yellow in Figure 5C,D represents the introduction of a hydrophobic group to facilitate an increase in the inhibitory activity of the compound, while the white region represents the introduction of a hydrophobic group, which reduces the inhibitory activity of the compound. The hydrophobic field map shown in Figure 5C demonstrates that the nitro group of the anthraquinone ring and the left side-chain of the anthraquinone ring have white regions, indicating that this group can enhance the hydrophobicity of the molecule and further enhance the antiproliferative activity of the compound. Figure 5D shows a hydrogen bond acceptor field for the CoMSIA model. It can be seen that the nitro group of the anthraquinone ring and the left side-chain of the anthraquinone ring are surrounded by a magenta region, with only a small red region around the anthracene ring, indicating that the hydrogen bond acceptor group at this position is advantageous.

### 2.4. 8a Induces Apoptosis in HCT116 Cells

Natural anthraquinones, such as rubiadin, physcion, emodin and soranjidiol [25,26,27], have been reported to induce apoptosis in tumor cells. Therefore, after detecting the effect of **8a** on tumor cell proliferation, we investigated correlation between the anticancer activity of compound **8a** and apoptosis. The effects of compound **8a** treatment on HCT116 cell apoptosis were detected by flow cytometry following Annexin V/PI staining. Treatment with compound **8a** at different concentrations (0, 10, 20, and 40 μg/mL) for 24 h significantly increased the rate of apoptosis (11.24%, 18.97%, and 26.94%, respectively) compared with that in the control group (0.06%) (Figure 6A,B). This indicated that the mechanism by which **8a** inhibits cell proliferation may associate with apoptosis.

In addition, the expression of apoptosis-related proteins after **8a** treatment was detected by western blot. Previous studies have shown that the Bcl**-**2 protein family is one of the most interesting oncogene-related proteins in programmed cell death [28]. Therefore, we analyzed the levels of Bcl**-**2, Bcl**-**xL and Bax after 24 h of treatment with different concentrations (0, 10, 20, and 40 μg/mL) **8a**. The results showed that the levels of Bcl**-**2 and Bcl**-**xL gradually decreased with increasing **8a** concentration, whereas Bax expression was increased. Moreover, the phosphorylation levels of Bcl**-**2 and Bcl**-**xL increased with increasing **8a** concentration. These results indicated that **8a** activated Bax expression and inhibited the expression of Bcl-2 and Bcl**-**xL protein (Figure 7A,C). This suggested that **8a** may induce apoptosis in HCT116 cells by regulating Bcl-2 family proteins.

Caspases represent a key molecular group in the regulation of apoptosis. In caspases, there is an upstream and downstream relationship between the initiator of apoptosis and executioner. Caspase 3 is one of the key executers of apoptosis, as it is either partially or totally responsible for the proteolytic cleavage of many key proteins. For example, when caspase 9 is activated, it activates caspase 3, which ultimately triggers cell apoptosis [29,30]. After detecting the expression of the corresponding Bcl**-**2 family proteins after **8a** treatment by western blotting, we went on to detect the expression levels of PARP, caspase 9, cleaved caspase 9, caspase 3, and cleaved caspase 3 proteins. As shown in Figure 7B,D, caspase 9 and caspase 3 expression levels in HCT116 cells gradually decreased with increasing of concentrations of in **8a**, while the expression of cleaved caspase 9 and cleaved caspase 3 increased in a dose-dependent manner. zVAD-FMK is a pan**-**caspase inhibitor that penetrates cell membranes and binds irreversibly to the catalytic site of caspase proteases, thereby inhibiting apoptosis. As shown in Figure 7E, HCT116 cells were co**-**incubated with **8a** (20 μg/mL) and zVAD**-**FMK (5 mM) for 24 h, after which cell viability was measured by MTT assay. After 24 h of treatment with zVAD**-**FMK, cell viability is 93%. However, co**-**incubation with **8a** and zVAD-FMK, cell viability increased significantly from 38% of the effect observed following treatment with **8a** to 55%. These results indicated that caspases are activated by **8a**.

### 2.5. Compound ***8a*** Induces Generation of ROS and Affects Mitochondrial Membrane Potential

Intracellular ROS are key factors in the activation of many signaling pathways, involved in a variety of biological functions including apoptosis and necrosis. High levels of ROS are capable of inducing tumor cell apoptosis [31]. We evaluated the ability of the compounds to induce ROS by flow cytometric analysis of the fluorescent probe DCFH**-**DA. Compound **8a** was found to increase ROS levels in a dose**-**dependent manner compared to the 0.01% DMSO**-**treated control (Figure 8A,B), indicating that **8a** induces the production of intracellular ROS. Next, we investigated ROS production before and after induction with 8a and NAC (N-acetyl-L-cysteine). The results indicated that NAC prevents **8a**-induced ROS production (Figure 8C,D).

Decreased mitochondrial membrane potential is a hallmark of early cell apoptosis, and a decrease in mitochondrial membrane potential leads to the release of certain mitochondrial proteins, such as cytochrome c, which mediate apoptosis. Therefore, we examined changes in mitochondrial membrane potential by flow cytometry. HCT116 cells were treated with **8a** (10 or 20 μg/mL) for 24 h and then stained with 2 μg/mL JC**-**1 for 30 min prior to flow cytometric analysis. Green /orange fluorescence ratio of JC-1, compare with control, is 200% and 494.4%, respectively (Figure 8E,F). The results showed compound **8a** decreased the mitochondrial membrane potential in a dose-dependent manner. During apoptosis, mitochondrial cytochrome C is released into the cytoplasm and triggers the activation cascade of caspase. HCT116 cells were collected after 8a treatment, cytoplasmic protein and mitochondrial protein were extracted. The levels of cytochrome c in the cytoplasm and mitochondria were detected by Western blotting. After treatment of HCT116 cells with **8a** (10 or 20 μg/mL) for 24 h, cytoplasmic levels of cytochrome c increased, while the levels in the mitochondria decreased. The results indicated that compound **8a** reduces the mitochondrial membrane potential, thus implicating the mitochondria in **8a-**induced apoptosis (Figure 8G,H). 

### 2.6. Compound ***8a***-induced ROS Production Is Associated with JNK Activation

In the ROS-induced signaling pathway, JNK is an important downstream molecule that activates apoptosis. ROS induces JNK activation, which in turn induces the expression of pro-apoptotic proteins in tumor cells to activate apoptosis via the mitochondrial pathway [32,33]. JNK is a member of the mitogen-activated protein kinase family and is capable of phosphorylating and activating c-Jun. JNK is involved in regulation of biological processes, such as cell proliferating, cell differentiation, cell death and stress responses, and further promotes expression of p53, Bax, FasL and tumor necrosis factor by promoting the activity of transcription factor complex activator protein-1. 

To further elucidate the mechanism by which **8a** inhibits proliferation and induces apoptosis of HCT116 cells, the levels of phosphorylated and total JNK protein were examined by western blot analysis. As shown in Figure 9A, total JNK levels were not altered by a treatment with compound **8a**, while the levels of phosphorylated JNK gradually increased with time. Phosphorylation of JNK activated by the ROS pathway results in overexpression of tumor suppressor factors, thereby inducing apoptosis [34]. In addition, JNK phosphorylation is associated with ROS production [35,36]. To determine whether ROS production contributes to JNK1/2 phosphorylation and its related mechanisms, we blocked **8a**-induced ROS production in HCT116 cells and then analyzed the changes in JNK 1/2 phosphorylation by western blotting. As show in the Figure 9B,C, the results show that the levels of p-JNK after treatment of HCT-116 cells with NAC were on the level of control samples. Furthermore, pre-treatment of cells with NAC reduced the levels of p-JNK induced by 8a. These results indicate that compound **8a**-induced ROS production is associated with JNK activation.

Furthermore, to further investigate whether activation of phosphorylated JNK affects cell viability, HCT116 was co**-**treated with **8a** and SP600125 (JNK inhibitor), and the effect of phosphorylated JNK inactivation on HCT116 cell activity was examined by MTT. Compared with the control, HCT116 cell viability was significantly inhibited (39%) following exposure to **8a** (Figure 9D). Following co**-**treatment with **8a** and SP600126, the rate cell viability was 55%. The inhibitory proliferation activty of 8a were partially weakened by the usage of 5 μM Sp600125. The results showed that the increase of the expression of phosphorylated JNK after 8a treated will decrease the cell viability. It also indicated that the increase of the expression of phosphorylated JNK will inhibit cell proliferation, which is related. 

## 3. Discussion

Natural anthraquinone compounds have been shown to exert anticancer effects both in vivo and in vitro [37]. Anthraquinone compounds induce cell death in a variety of cancer cell lines, including LS1034, HepG2, MCF**-**7, MDA**-**MB**-**231, MCF**-**10A, SW480, and SW620 cells [38,39]. Chemical modifications of the side-chain represent an effective approach to improving the activity of certain anticancer drugs.

Most of all the compounds inhibited cancer cell proliferation according to the MTT assay. Compounds in which the side**-**chain groups were replaced by leucine exhibited higher activity. Therefore, it is shown that **8a** has significant anti-proliferative activity on a variety of tested cells, and the effect on HCT116 cells is more significant. Furthermore, **8a** induced HCT116 cell line death via apoptosis in a time**-**and dose**-**dependent manners.

Anthraquinone induces the generation of ROS [40], which induce apoptosis of tumor cells via JNK activation [41,42]. Activated JNK phosphorylates Bcl**-**2 [43]. The production of ROS causes JNK activation and induces the death of various cell lines, such as human pancreatic cancer cells, Sertoli cells, and human cervical cancer cells [44]. Extracellular signal-regulated kinase 1/2 (ERK1/2), c-Jun N-terminal kinase (JNK), and p38 MAPK are the most widely studied members of the MAPK family [45]. Our data demonstrated that **8a** induced HCT116 cell apoptosis via the ROS/JNK signaling pathway. This finding suggests that ROS acts an upstream signaling molecule in the pathway. Treatment of HCT116 cells with **8a** increased ROS levels in a dose-dependent manner. In addition, inhibition of ROS generation by NAC (5 mM) markedly reduced activation of JNK in HCT116 cells, suggesting that **8a** induces HCT116 cell death, at least partially, through ROS-mediated activation of JNK. However, NAC only partially blocks 8a induced ROS production (Figure 8D). Some studies have shown that NAC has a concentration-dependent inhibitory effect on ROS induction [46]. Moreover, other studies have shown that NAC is a weak reducing agent and a poorer antioxidant compared to glutathione (reduced form) (GSH) [47], so the experimental results show that NAC partially blocks **8a**-induced ROS production. Moreover, we also concluded that although we have performed many preliminary experiments in the previous period, the concentration of 5 mM NAC still cannot quickly clear the ROS induced by **8a**. In addition, JNK inhibition only partially rescue the anti-proliferative effects induced by compound **8a** (Figure 9E), we concluded that **8a** inhibition of HCT1116 cell proliferation may also be related to the regulation of other signaling pathways.

In addition, using 3D-QSAR analysis, results indicated that if the hydrophilic target protein is near to the **8a** leucine side chain, it may increase antitumor activity and further induce apoptosis. In the next, it is an interesting and worthy further study of **8a**-induced apoptosis signaling pathways. The reduction in Bcl**-**2/Bax ratio was determined as a marker of an apoptosis signaling [48]. Numerous studies have shown that phosphorylation of Bcl**-**xL(S62), but not Bcl**-**2, induces cell apoptosis via Bax oligomerization in the mitochondria and cytochrome c release in the cytoplasm [22,49]. However, the results showed that both Bcl-2 and Bcl-xL were phosphorylated after treatment with **8a**, therefore, the specific mechanism of **8a**-induced HCT116 cell apoptosis needs further study.

## 4. Materials and Methods

### 4.1. Reagents

The following reagents and equipment were used: analytically pure (AR) 2**-**methylanthraquinone (Chemical & Technology Company, Shanghai, China); SHZ-C type circulating water multi**-**purpose vacuum pump (Gongyi City Yingyu Yuhua Instrument Factory, Henan, China), RE-52A rotary evaporator (Yarong Biochemical Instrument Factory, Shanghai, China); X-4 digital display micro-melting point tester (Beijing Tektronix Instrument Company, Beijing, China), HP8453 UV-Vis absorption spectrometer (Hewlett Packard, California, The United States); DRX300MHZ nuclear magnetic resonance instrument (Bruker, Worcester, Switzerland); RPMI-1640 medium (HyClone, Logan, Utah, The United States); fetal bovine serum (Sangon Biotechnology, Shanghai, China); MTT (Sigma, Maryland, The United States); Annexin V-FITC assay kit (Pharmingen Becton Dickinson, New York, The United States); anti-Bcl-2, anti-phosphorylated Bcl-2 (Bioss, city, China); anti-Bcl-XL, anti-phosphorylated Bcl-XL, anti-Bax, anti-Caspase-3, anti-Caspase-9, anti-cleaved caspase-3, anti-cleaved caspase-9, anti-PARP antibody, anti-phosphorylated (Thr183/Tyr185) JNK, anti-JNK, ROS detection kit, N-Acetyl-L-cysteine (NAC) and mitochondrial membrane potential detection kit (Beyotime Biotechnology, Shanghai, China); anti-cytochrome c, anti-COX**-**IV antibody, horseradish peroxidase-labeled goat anti-rabbit IgG secondary antibody, anti-β-actin antibody (Bioworld, Minnesota, The United States); ECL assay kit (Engreen Biosystem, Beijing, China); PVDF membrane (Millipore, Massachusetts, Germany); BCA protein quantification kit (Minbio, Shanxi, China).

### 4.2. Compounds

We used 2-methylanthraquinone as a raw material to synthesize the 1-nitro-2-benzoyl chloride anthraquinone intermediate by sequential nitration, oxidation, and acylation. The 1-nitro-2-acylanthraquinone-amino acid derivatives **8a**–**8h** were then synthesized by reaction with suitable amino acids. The characterization results of *N*-[(9,10-dihydro-1-nitro-9,10-dioxo-2-anthracenyl)carbonyl-leucine] (**8a**) are as follows: IR, v/cm^-1^: 3487 (w), 3371 (w), 3500–2500 (w), 2960 (w), 1724 (m), 1680 (s), 1658 (s), 1589 (m), 1552 (s), 1469 (w), 1411 (w), 1369 (w), 1319 (m), 1280 (s), 711 (m); ^1^H-NMR (300 MHz, DMSO), δ ppm: 0.85–0.91 (m, 6H), 1.55–1.70 (m, 3H), 4.34 (s, 1H), 7.91–7.94 (m, 2H), 8.01–8.20 (m, 3H), 8.44–8.47 (d, 1H), 9.29–9.31 (d, 1H), 12.82 (s, 1H); ^13^C-NMR (1200 MHz, DMSO), δ ppm: 23.6, 25.5, 26.9, 42.3, 53.4, 129.4, 131.4, 135.1, 135.6, 136.7, 137.5, 138.0, 148.5, 165.7, 175.9, 182.4, 183.1; UV λ max (THF) (nm): 261, 326. The characterization results of the other anthraquinone derivatives **8b–8h** are given in the Appendix A.

### 4.3. Cell Culture

The human esophageal cancer cell line EC9706 was obtained from Cancer Hospital and Institute by Prof. Ming Rong Wang, Chinese Academy Medical of Sciences. The human colorectal carcinoma cell lines HCT116 and SW480, hepatoma cell line HepG2, cervical carcinoma cell line HeLa, breast cancer cell line MCF-7, gastric cancer cell line SGC-7901, cholangiocarcinoma cell line QBC939, normal fetal colon cell line FHC and liver cell line HL-7702 were obtained from the Cell Bank of the Shanghai Chinese Academy of Sciences. All cells were cultured in RPMI 1640 medium containing 100 U/mL penicillin and streptomycin and 10% fetal bovine serum and maintained at 37°C under a 5% CO_2_ atmosphere.

### 4.4. In Vitro Cell Proliferation Assays

We analyzed the inhibitory effects of eight anthraquinone-amino acid derivatives on the proliferation of eight tumor cell lines and two normal cell lines in vitro using MTT assays. Cells were added to a 96-well plate with (5,000 cells per well in 100 µL RPMI-1640 medium). Cells were treated with different concentrations (10–100 μg/mL) of **8a**–**8h** for 48 h and then incubated with 20 μL MTT (5 mg/mL) for another 4 h at 37°C. DMSO (150 μL) was then added to each well to dissolve the formazan crystals. The absorbance of each well was read at 490 nm on a microtiter plate ELISA reader. The mean values were obtained from a minimum of three parallel experiments. IC_50_ values were calculated.

### 4.5. 3D-QSAR Study

The amide anthraquinone derivatives (**8a**–**8h**) were subjected to 3D-QSAR studies using Comparative Molecular Field Analysis (CoMFA) and Comparative Molecular Similarity Index Analysis (CoMSIA), which showed inhibition of the proliferative activity of colon cancer cell lines. Since the antitumor activity of **8a** was the highest, the structure of Compound **8a** was used as a representative for analysis. The CoMFA and CoMSIA models were generated using the SYBYL-X 2.0 QSAR software module, in which the negative logarithm of the IC (pIC_50_) was used as the modeling response value. The IC_50_ value was measured in terms of colon cancer cell proliferation. Structural changes (CoMFA fields) were correlated with changes in the antiproliferative activity of the amide anthraquinone derivatives using partial least squares regression (PLS).

### 4.6. Apoptotic Assay

HCT116 cells were added to a 6-well plate (5 × 10^6^/well). After attachment, the cells were treated with **8a** for 24 h. Then, the cells were collected and centrifuged (10,000 g × 5 min). Cells were washed once with 1 mL of PBS. Cells were suspended in 500 μL binding buffer (containing 5 μL Annexin V-FITC and 1 μL propidium iodide [PI]) and incubated for 30 min in the dark. Apoptosis cells were detected by flow cytometry (FACSCalibur, BD, The United States).

### 4.7. Detection of ROS by DCFH-DA

Cells were added to a 6-well plate (1 × 10^4^/well) and incubated for 24 h to allow adherence. Set 0.1% DMSO as the control group, 8a as the treatment group, NAC as the treatment group, 8a + NAC as the treatment group. Among them, in the 8a + NAC treatment group, 8a was added after 5 mM NAC treatment for 1 h. Cells that had been treated for 24 h were collected. The cells were then washed three times with PBS and incubated with 10 μM DCFH-DA for 30 min in the dark. ROS levels were analyzed by flow cytometry.

### 4.8. Determination of Mitochondrial Membrane Potential

The 1×10^4^ cells were treated with different concentrations of **8a** for 24 h. Cells were then incubated with 500 μL JC-1 (10 mg/mL) for 30 min at 37 °C in the dark. The level of mitochondrial membrane potential depolarization was measured by flow cytometry. Data were analyzed using the Cell Quest program (BD, New York, The United States).

### 4.9. Western Blot Analysis

The expression of signaling pathway-related proteins was detected by western blot analysis. Cell (1 × 10^6^) were plated in 90-mm dishes and treated with 8a for 24 h. The cells were treated with NAC in the same way as in step 4.7. Resuspend the cells in Buffer A (1 mM EDTA, 1 mM PMSF, 0.28 μg/mL apotinin, 50 μg/mL leupeptin, 7 μg/mL pepstain A) and transferred to a 1 mL glass homogenizer and pull up and down 30 times in an ice bath. Cells were collected and centrifuged (1000 g × 10 min, 4 °C). The obtained supernatant and precipitate were the crude cytoplasm and mitochondria, respectively. The cytosolic fraction was transferred to an ultra-ion tube and centrifuged (100,000*g* × 1 hr, 4 °C), and the resulting supernatant was the purified cytosolic fraction. The mitochondrial pellet was resuspended in Buffer B (1mM EGTA, 1mM PMSF, 0.28 μg/mL apotinin, 50 μg/mL leupeptin, 7 μg/mL pepstain A), centrifuged (10,000*g* × 10 min, 4 °C), and repeated 3 times. The mitochondria pellet was lysed in cell lysate buffer (1% NP-40, 2 μg/mL aprotinin, 2 μg/mL leupeptin, 1mM EDTA, 1mM Na vanadate). Protein concentrations in the supernatant were determined by using a Minbio BCA protein assay kit. Protein were separated by SDS-PAGE and transferred to a PVDF membrane. The membrane was blocked, and incubated with primary detection antibodies overnight at 4 °C. After washing three times with TBS-T, the membranes were incubated with the secondary detection antibody. After washing three times with TBS-T, protein bands were visualized using an enhanced chemoluminescence (ECL) kit (luminol/enhancer solution, peroxide solution, protein detection complex). Next, in the gel imaging system, the protein bands were scanned and the pictures were imported into the Image J software for gray analysis. 

We have adopted two analysis methods:
Relative protein level/fold = Experimental group gray value/ Control group gray value
Ratio of relative intensity of control = Experimental group relative protein level/Control relative protein level

### 4.10. Statistical Analysis

All data in this paper were performed in three independent experiments. The SPSS 11.5 software (company, city, state abbrev if USA, country) was used to calculate IC_50_ values. Differences among groups were assessed using one-way analysis of variance (ANOVA). * *p* < 0.05, ** *p* < 0.01 indicates a statistically significant difference.

## 5. Conclusions

In summary, we have generated a series of new anthraquinones, compounds **8a-8h**, by structural modification of anthraquinone, an active ingredient of several traditional Chinese medicines. Among the compounds generated, our study showed that **8a** has the best anti**-**tumor activity against colon cancer cells and induces apoptosis via the ROS/JNK signaling pathways. In **8a**-induced apoptosis, initial activation of the ROS-JNK signaling pathway caused increased ROS production, JNK phosphorylation, followed by a decrease in mitochondrial membrane potential and release of cytochrome c mediated by the actions of Bax and Bcl**-**2. These results lead to caspase 9 and caspase 3 cleavage, which in turn leads to activation of the caspase signaling pathway, and ultimately to apoptosis (Figure 10). Moreover, by analyzing structure**-**activity relationships, we have demonstrated that the solubility and biological activity of the compound synthesized can be greatly improved by increasing the electron withdrawing capacity of the nitro group at the C**-**1 position and substitution of the amino acid at the β position. Thus, our study provides new insights for the future design and development of more effective anti**-**tumor drugs.

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
