# Peer review of "Novel Anthraquinone Compounds Inhibit Colon Cancer Cell Proliferation via the Reactive Oxygen Species/JNK Pathway"

_molecules, 2020, doi:10.3390/molecules25071672_

Round 1
Reviewer 1 Report
I would like to thank the authors to the work realized to improve their study. They well answered to the majority of my questions.
However, some mistakes and corrections should be edited before possibility to be published:
- Authors should homogenize the y-axis of the graphs corresponding to western blot quantifications
- Figure 7 panel C / Figure 8 panel H / Figure 9 panel D: authors should exchange radio by ratio.
- The figures’ quality is less good than the previous version. For instance, it may be interesting to increase the size of the histograms in the panels C and D from the figure 7.
- Figure 8: the two axis of the panel E are similar. However for JC-1 assay, you should record green channel and red channel. The authors must modify this point.
- Figure 8 panel F: authors should modify y-axis in Mitochondrial Membrane Potential (% of…)
- Line 298: authors described the level of total Jnk in the panel C of figure 9. However, the panel C represents the level of p-Jnk. Authors should check this point.
- Lines 304 to 308: information reported by the authors are not in accordance with the results presented in the figure. NAC exposition doesn’t reduce the p-JNK level compared to the control and the simultaneous application of NAC with 8a doesn’t drive a reduction of p-JNK compared to the 8a condition. After a comparison with the first version of the work, it appears that this text is corresponding to this first version. The authors should check the entire text of the article.
- Line 406: a space is lacking between the figure and the unit.
- Line 453: Navanadate should be replace by Na-vanadate
Reviewer 2 Report
In this manuscript the authors generate anthraquinone derivatives and perform a study with this compounds aimed to determine their efficacy as antitumorigenic drugs. Previous studies have established the antitumor activity of several antrhaquinone derivatives. However, various problems, including toxic side-effects or poor solubility, limit the use of these drugs in the clinic. To solve these problems, the authors combine several amino acids to 2-methylanthraquinone to generate eight different anthraquinone-amino acid derivatives. To evaluate the antitumorigenic effects of these compounds the authors evaluate cell proliferation and apoptosis and made use of computational research (structural function relationship 3D-QSAR) to predict structure-function relationship. These studies identify compound 8a as the most potent among the anthraquinone derivatives generated and determine that this compound inhibits proliferation of HCT116 colon cancer cells by inducing apoptosis. The authors also try to determine the molecular mechanisms underlying the pro-apoptotic action of compound 8a by evaluating JNK activity, ROS generation and changes in mitochondrial membrane potential. Based on the results of these studies, the authors conclude that compound 8a induces apoptosis by inducing ROS production that in turns activate JNK, which induces mitochondrial stress, release of citochorme c and activation of caspases.
The study is interesting, provides valuable information for those working on cancer and has potential clinical relevance. However, there are some major points that should be addressed before considering the work for publication:
Major points:
- The statement that ROS production by 8a results on JNK activation is not properly demonstrated. First of all, title in section 2.5 (Compound 8a induces JNK activation via ROS generation, line 249) should be changed since the results on this section demonstrate that 8a induces generation of ROS and affects mitochondrial membrane potential, but JNK activation is not shown here. More importantly, although results in figure 9 clearly show JNK activation after 8a treatment, incubation of cells with NAC, which blocks ROS production, clearly stimulates JNK activation either alone or in combination with 8a compound (Fig 9B). Therefore, inhibition of ROS production by 8a after NAC treatment does not block JNK activation. Authors should either demonstrate that JNK is activated by ROS production after 8a treatment or eliminate these results from the paper, and therefore, modify conclusions accordingly.
- Title in section 2.6 (JNK activation is involved in the changes in expression levels of apoptosis-related proteins induced by 8a treatment, lines 286-287) should be changed. There is no analysis of apoptosis-related proteins in this section. Additionally, results in figure 9E show that JNK inhibition only partially rescue the anti-proliferative effects induced by 8a compound. Therefore, this result does not fully support that 8a induces apoptosis through JNK activation. The authors should indicate that this is a partial contribution. Furthermore, an experiment showing that inhibition of JNK by SP600125 rescue the pro-apoptotic effect of 8a (by apoptosis analysis) should be included, since measurement of proliferation does not directelly evaluate apoptosis.
Additional points
- In table 1 standard deviation should be included in each IC50 value since, as indicated in Materials and methods (section 4.4), experiments were performed at least three times.
- How reproducible are the effects of 8a? For example, in fig 8d, the effect of 8a on ROS production is much more potent that in 8b. Similarly, in fig 9a, the effect on JNK phosphorylation is much clear than in fig 9b.
- The number of experiments performed for fig 7E should be included.
- Results of Fig. 7E are not properly explained in the text (line 235 and 236). It makes no sense to state that zVAD-FMK treatment increases cell viability to 93% since cell viability of cells treated with this compound is similar to that of control.
- Discussion is too general and several aspects of the study are not discussed. For example, the fact that NAC only partially blocks 8a ROS production (Fig 8D) and that JNK inhibition only partially rescue the anti-proliferative effects induced by 8a compound (Fig 9E) should be discussed. In general, the work should be more deeply analyzed in the context of previous works (only 9 references are cited in discussion).
- Title should be modified. Since most of the experiments have been performed only in HCT116 cells, the title should indicated that the anthraquinone effects are restricted to this cell line or, at least, to colon cancer cells.
- Line 98; please include also alanine, methionine and glutamic.
- Line 102; add hyphen (change SGC7901 to SGC-7901).
- Title of Tables 2 and 3: include in HCT116 cells.
- Line 221, eliminate **p<0.01 since there are no results with that p value.
- Legend of Fig. 8G. β-actin is a marker of cytosolic fraction and COX-IV of mitochondria. Please correct.
- Line 296. …8a inhibits proliferation and induces apoptosis of HCT116 (include induces).
- English usage needs attention. There are various typos and there are also sentences that are difficult to understand. As examples:
- Line 119; change larger for higher
- Sentences from line 339 to line 344 (According to…) are difficult to understand.
- The number of cells with a low decrease in mitochondrial… (line 263). What does it means?
- Include that after The results showed in line 264
- An introductory sentence is needed before describing the analysis of cytochrome c (before HCT116 cells were collected… line 265).
- Legend of Fig 8D is unclear.
Reviewer 3 Report
The study of Li et al. presents research regarding the cytotoxic and pro-apoptotic activity of anthraquinone derivatives. The results of their study showed moderate cytotoxic activity of the examined compounds, with IC50 values ranging from 21 to 73 µg/ml. The highest activity was that of a 1-nitro-2-acyl anthraquinone-leucine, with an IC50 value of 17.8 µg/ml toward HCT116 cells. This compound was found to induce apoptosis and ROS in HCT116 cells. Apoptosis was determined to be induced through ROS-mediated JNK activation, triggering the mitochondrial cell death pathway.
The authors undertook an interesting topic related to identifying novel anthraquinone derivatives with potential anti-cancer activity. Below are my comments regarding this research:
Major points:
It would be more informative to compare the cytotoxic activity of compounds in their µM concentrations.
An IC50 value of 21 µg/ml for a compound is not strong cytotoxic activity. Furthermore, the observed cytotoxic activity of compounds in the range of 21 to 73 µg/ml is not ‘significant’ cytotoxic activity but moderate.
Please provide the procedure for treatment of cells with NAC in the Materials and Methods section, including time of treatment, concentration etc.
The results related to the effects of 8a on JNK activation described in the results section do not correlate with the those presented in the figure. In figure 9B and D, we can observe an increase in p-JNK levels upon NAC treatment. Furthermore, the level of NAC+8a are at the level of NAC treatment, which is inconsistent with the graph in Figure 9D. From what is presented in Figure 9B, it seems as though the selected concentration of NAC was too high as it increased p-JNK and therefore would not be applicable to this experiment.
In the discussion section, the paragraph related to apoptosis induction by 8a should be re-written. The authors claim in the discussion that 8a did not induce apoptosis.
Please check the Experimental section regarding Annexin V staining. This procedure cannot be performed after fixing cells with 70% ethanol.
Minor points:
-please change sentence in line 257 to: NAC prevents 8a-induced ROS production.
Round 2
Reviewer 1 Report
Authors answered correctly to the different points and improved their work to obtain publication.
Author Response
We have revised all your suggestions, and thank you for your suggestions.
Reviewer 3 Report
After the revision of the authors answers I still have some comments:
Line 340: ‘ According to the results of flow cytometry, after 8a treatment of HCT116 cells, it appeared that the number of apoptosis was relatively less. The possible reason is that apoptosis occurs in the early stages of cell death’. I still do not understand this conclusion. Flow cytometry analysis showed a dose-dependent increase in apoptotic cells, at lower concentrations a higher increase in early apoptotic cells was visible, whereas at higher concentrations, late apoptotic cells were more prevalent.
Line 354: NAC is a ROS scavenger so the statement that ‘NAC does not inhibit ROS production’ is not accurate. The authors would have to experiment with various concentrations of NAC in order to obtain a greater decrease in ROS levels.
Line 307: ‘p-JNK levels decreased after treatment with NAC compared with the levels detected in the 8a’. The levels of p-JNK in cells treated with only NAC did not decrease in comparison with levels of p-JNK in cells treated with 8a. It would be more clear to write: The results show that the levels of p-JNK after treatment of HCT-116 cells with NAC were on the level of control samples. Furthermore, pre-treatment of cells with NAC reduced the levels of p-JNK induced by 8a.
Author Response
Line 340: ‘ According to the results of flow cytometry, after 8a treatment of HCT116 cells, it appeared that the number of apoptosis was relatively less. The possible reason is that apoptosis occurs in the early stages of cell death’. I still do not understand this conclusion. Flow cytometry analysis showed a dose-dependent increase in apoptotic cells, at lower concentrations a higher increase in early apoptotic cells was visible, whereas at higher concentrations, late apoptotic cells were more prevalent.
Response:We have modified here.
Line 354: NAC is a ROS scavenger so the statement that ‘NAC does not inhibit ROS production’ is not accurate. The authors would have to experiment with various concentrations of NAC in order to obtain a greater decrease in ROS levels.
Response:We have modified here.
Line 307: ‘p-JNK levels decreased after treatment with NAC compared with the levels detected in the 8a’. The levels of p-JNK in cells treated with only NAC did not decrease in comparison with levels of p-JNK in cells treated with 8a. It would be more clear to write: The results show that the levels of p-JNK after treatment of HCT-116 cells with NAC were on the level of control samples. Furthermore, pre-treatment of cells with NAC reduced the levels of p-JNK induced by 8a.
Response:We have modified here.
This manuscript is a resubmission of an earlier submission. The following is a list of the peer review reports and author responses from that submission.
Round 1
Reviewer 1 Report
The manuscript presented by Authors entitled “Novel Anthraquinone Compound Inhibits Cancer Cell Proliferation via the Reactive Oxygen Species/JNK Pathway” shows anticancer properties of different anthraquinone derivatives. Authors synthesized a series of novel amide anthraquinone-amino acid derivatives using 2-methylanthraquinone as a raw material, studied their cytotoxicity, performed 3D-QSAR to correlate biological activity with the chemical structures. As cellular models for biological studies eight human cell lines with different tissue origin were used. Authors also explored influence of synthesized compounds on metabolic activity against two normal cell lines. Presented results showed that among eight of studied compounds compound 8a revealed proapoptotic properties in HCT116 cancer cells. I have to emphasize that presented manuscript reveals tremendous work of the Authors, who not only synthesized new derivatives and showed their spectroscopic characterization, but also tried to explain the molecular mechanism of apoptosis induction by 8a.
The results presented are relevant and the strategy is well established. The bibliography is up-to-date. I recommend the acceptance of the manuscript to be published in Molecules after some revisions.
During in vitro studies of biological activity of pure chemical compounds usually there is used µM unit to describe effective concentration, whereas µg/mL is used for extracts; thus I suggest to change the units in the manuscript;
Do the Authors performed comparable in vitro studies for parent compound to compare the biological effectiveness of synthesized derivatives?
Comparison of IC50 dosages shows that compounds 8a-8e possess comparable biological activity; why compound 8a was chosen for more detailed study; do the Authors performed comparable studies with compounds 8b-8e?
Line 31 - There is no dot at the end of sentence; unify through the entire text;
Line 31 – change of the “synthesize” into “synthesis” should be done;
Line 32 – the “act” word is repeated;
Lines 32-34 – change the structure of sentence;
Lines 35-36 – anticancer and antitumor activities are comparable activities, thus the sentences should be modified according to the presented information;
Lines 29-63; 141-152; 159-166; 170-176, Figure 8; Figure 10; 401-419 – size of font should be changed;
Line 63 – change “caspase” into “caspases”;
In Table 1 there should be value 342.19 instead of 342.191; however it is not necessary to present data with two decimal places; still I suggest to present data in µM;
In Table 2 there are no results for FHC cell line;
Table 3 should be edited;
Figure 6 B – there should be “apoptotic” instead of “apoptosis” in Y axis title;
Line 209 – there is no explanation about checking the 8a compound involvement in apoptosis induction at 20 µg/mL; please present more information in that field;
Lines 232-233 – The structures of sentences should be changed;
Line 257 – Authors wrote that checked the influence of 8a on MMP at 10-20 µg/mL dosages, whereas the identification of mechanism od apoptosis induction was performed for 20 µg/mL; could you explain that?
The Authors showed results connected with apoptosis induction, whereas dosages used for study were higher than IC50 (17.80µg/mL); did the Authors observe necrosis induction in studied cells?
Figure 9 – why COX IV was used as internal control (with β-actin) in Figure 9, while it is not presented in Figures 7 and 10?
Line 381 – the concentration of MTT should be written;
Line 399 – change “apoptosis” into “apoptotic”;
Lines 432-433 – The structures of sentences should be changed;
References should be edited and unified.
Reviewer 2 Report
In this work, Li et al.described the effect of a new family of anthraquinone derivatives on the cancer cell proliferation. Their results are very interesting and they highlighted the compound 8a as a new lead in their series.
Regarding the chemistry synthesis and analysis, I don’t possess the background necessary to evaluate correctly their work. However, authors were very clear and well justified their choice of the compound 8a (based on the comparison of the other modified compounds). In addition, they provided information about the structure function relationship of their lead and cell proliferation inhibition.
The mechanistic analysis requires more improvements. Although the authors deciphered partially the signaling pathway involved in the answer to the 8a exposition, it appears necessary to improve the presentation of their results. Indeed, statistical analyses are not in adequation with the current standard and some experiments are duplicated without justification. In addition, multiple mistakes are detected all along the text. Finally, the readers are curious about the putative direct target of the compound, which is not at all discussed in the manuscript.
Major points:
- Authors should to check the full text to homogenize the presentation of their results. More precisely, I noted multiple mistakes in the legends (the description is not in accordance with the described panel). Cytometry results in the figure 8 and 9 are overlapped and it is not necessary to duplicate these sets of results.
- Section 3.10: the statistical analysis section is not correct.
In addition, the scientific rigor is not demonstrated by the use of t test. Indeed, when the authors compare more than two means, they absolutely must use ANOVA assay (or relative non parametric). All along the main text, the authors don’t precise what they apply to analyse their results.
In the method section, authors wrote, “All data in this paper were performed in three or more independent experiments.” However, they indicated in all legends that data are extracted from 3 independent experimments. They should modify the method section or they indicated all the replicate of their experiments.
- Authors could discuss the putative direct target of their compound in addition to the description of the apoptosis’ signaling pathway.
Minor points:
- Lines 31-32: “Therefore, the design and synthesize of new compounds that 32 act specifically act on target proteins in tumor cells is a focus of research” should be replace by “Therefore, the design and synthesize of new compounds that 32 act specifically act on target proteins in tumor cells is a focus of research.”
- Lines 35-42: add a point at the end of the sentence.
- Figure 1 - legend: authors should add a comment about the red section in each compound.
- Lines 58-63: authors should move the reference at the end of the sentences to homogenize with the rest of the text.
- Lines 64-66: the authors should check the concept of mitochondrial apoptosis sequence. First, it is a depolarization of the membrane followed by the release of cytochrome c. The authors should reformulate their description.
- Tables 1 and 2: it should be interesting to condensate the two tables in only one.
- Line 109 / table 2: there is a layout problem for the last 2 columns
- Line 141: “we conducted a 3D-QSAR study a series of” should be replace by “we conducted a 3D-QSAR study to a series of”
- Table 3: authors should to check the presentation of the data in the table and adjust it.
- Figure 3 - legend: what are the meanings of the panel A and of the panel B? It should be add in the legend.
- line 200: % is lacking after the second value.
- Figure 6B: it appears relatively few apoptosis compared to classical results of this assay in the control condition. In addition, what is the meanings of ** on the control bar? This last comment is also applicable in figure 7 (panels C and D), figure 8 (panels B and C).
- Figure 6 - legend: replace Date by data. It is also applicable to figures 7(E), 8(B,C), 9(B,C) and 10.
- Lines 218-220: authors should check the sentence.
- Line 223: authors wrote “cleaved caspase X”. In the following section, they indicate cleave caspase X. They should check it and correct in cleaved.
- Figure 7a: first, could the authors add the reference of the anti-p-Bcl-2 antibody? Second, it is not classical in the literature that the p-Bcl-2 has the same molecular weight to Bcl-2? In addition, it is relatively curious that you obtain more intensity in the phospho-condition than in the total Bcl-2 protein.
- Figure 7c: normalization makes reading easy. It should be interesting to normalize the ratio to the control.
- Figure 7 - legend: the legend does not correspond to the different panel of the figure.
- Lines 230-233: authors should reformulate the presentation of the results from the figure 7e.
- Line 251: “dose-dependnet” should be replace by dose-dependent
- Line 259: the values presented in the text don’t correspond to the values shown in the figure. In addition, authors presents 3 values but they evaluated two concentrations in comparison to control.
- Figure 8: the organization of the figure should be revised to optimize the reading of the information. Panel b should be move to the right of the panel A and panel C and D should be placed similarly.
- Figure 8: the labels of the graphs C and D should be check. In panel C, the authors should add potential and in the panel D, in place of the name of the channel, authors should precise the parameter recorded.
- Figure 8 - legend: the description does not correspond to the according panel.
- Line 271: authors described the distribution of the cytochrome c in cells treated with 8a. They noted that with the 8a concentration increase, the mitochondrial fraction decreased and, inversely, the cytoplasmic fraction increased. However, it is never notice how the authors could identify the mitochondrial fraction in comparison to the cytoplasmic fraction. I strongly suggest that the authors add a paragraph in method section to explain how they prepare the cell fraction. In addition, the corresponding histogram presents surprising results. Indeed, in the panel 9b, we observed that the cytoplasmic fraction of cytochrome c decreases similarly to the mitochondrial fraction.
- Lines 276-277: authors wrote “Next, we investigated the influence of 8a-induced ROS production on JNK activation.” This affirmation is false. In this set of experiments, they only analysed the 8a-induced ROS production. Authors should adapt the text.
- Lines 277-279: “Cells were treated with 8a for 24 h with or without the addition of the ROS inhibitor NAC (1 nM) and stained with DCFH-DA prior to flow cytometric analysis. The results indicated that NAC prevents ROS production (Figure 9C, D).” This sentence is not necessary because the authors do similar experiments than previously.
- Figure 9: Panels C and D present similar results to figure 8 panels A and B. It is not necessary to duplicate experiments. The single addition obtained in this experiment is the confirmation of ROS production by the use of NAC (abbreviation that was not explained in the main text).
- Line 310: “with the levels deteced in the control.” should be replace by “with the levels detected in the control.”
- Line 311: “when the cells were treared” should be replace by “when the cells were treated”
- Line 318: in the text authors said “In contrast, HCT116 cell viability maintained at 95% of that detected in the control following treatment with the JNK inhibitor SP600126.” However, authors present a statistic difference in the corresponding graph. It should be important to improve the representation of the statistic analysis.
-Lines 391-321: “Following co-treatment with 8a and SP600126, the rate cell viability was 55%, which was significantly higher than that of HCT116 cells treated with 8a alone.” Based on the information about the statistic analysis used in the method section, we could not agree with this affirmation. The authors should use ANOVA (or related non parametric assay) to conclude in this way.
- Figure 10: In the panel A, authors showed that in control condition, there is no p-JNK while in the panel B, authors presented a relatively important detection of p-JNK. How authors explained this difference without information in the main text and in the legend?
In the panel B, the line around the p-JNK blot is not continue.
- Figure 10 - legend: The authors indicated in the panel B description that they studied the JNK expression but it is lacking in the corresponding figure.
- Line 346: “anti-cleave anti-caspase-3” should be replace by “anti-cleave caspase-3”
- Line 348: “cytochrome c” should be replace by “anti-cytochrome c”
- Line 396 - Section 3.6: authors should add details to the methodology (like cell number…)
- Lines 411- 412: what is the lysis buffer? rpm is not an international system unit.
- Line 416: What is the system used to detect the luminescence?
- Line 429: “Thus, 8a was shown to exert notable antiproliferative activity on HCT116 cell.”. Based on the data presented in the tables 1 and 2, the compound 8a seems to be efficient on other cell lines. Authors should precise these effects.
- Lines 432-444: In this paragraph, authors well described the signaling pathway. However, they don’t discuss the target of their compounds. It should be interesting to discuss this point in the discussion section.
- Lines 451-453: Reformulate “These effects result in activation of the caspase signaling pathway is activated, causing caspase 9 and caspase 3 cleavage, and ultimately leading to apoptosis (Figure 11)”
- Figure 11: Homogenize the writing of cytochrome c with the main text.
- Figure 11 - legend: the authors should add the abbreviations used in the figure to make easy the reading.
- References section: authors should to homogenize the presentation of the references.
Round 2
Reviewer 1 Report
The Authors of the manuscript presented “Novel Anthraquinone Compound Inhibits Cancer Cell Proliferation via the Reactive Oxygen Species/JNK Pathway” answeared all my questions sufficiently. Despite some editorial revisions needed (i.e. line 65) I recommend the acceptance of the revised manuscript to be published in Molecules.
Author Response
We have modified the format of 65 lines.
Reviewer 2 Report
Reviewer comments
I would like to thank the authors to the work provided in order to improve their study.
However, some points require more explanation to allow the publication.
Specific comments
First, authors answered that they adapted the duplicity of the results in figures 8 and 9 (answer 1). However they argue that the aims of the two experiments are not similar (answer 33). I think it would be wise to assemble the 2 figures into one. Indeed, the message of these two sets of experiments is in the same way.
Regarding the statistic analysis, I’m not agreeing with the authors. The justification to use T-test is not adequate. In their approach, the authors do not take into account the risk of error due to the multiplicity of tests. The pairwise significance analysis is not the good argument to select T-test in place of Anova analysis. The pairwise analysis is a requirement to assay the fact that data are paired or not. In the case of one parameter measured in multiple groups, you should to use Anova (or relative non parametric assay). One very good support to explain statistic analysis is: McDonald, J.H. 2014. Handbook of Biological Statistics (3rd ed.).
In this case, authors should to revise all the statistical analysis before publication.
Point 17: The authors don’t respond to the question about the weak mortality rate in their experiments. They should add comment about that in the text.
Point 18: The authors don’t do the adequate modification. They don’t replace date by data.
Point 19: the authors don’t modify the sentence. There is always a problem in this sentence: “In caspases, there is an upstream and downstream relationship between the initiator and of apoptosis and executioner molecules.”
Line 209: authors add a “d” at the end of the caspase in place of cleave.
Point 21: The authors should add the quantitative analysis method in the method section.
Point 22: the authors don’t answer to the proposition. In addition, they add a comment about their statistic analysis, which is always inadequate to the assay.
Lines 223-225-246-249-301-303: authors used date in place of data. Please modify this point.
Figure 7 panels C and D: the authors add * on the control histogram. What is the meaning of this addition?
Figure 8: in the panel D, authors didn’t change the axis names as previously requested
Point 30: the authors didn’t answer to the question about the cellular fraction. How could they identify mitochondrial fraction compared to the cytosolic fraction? They should add information in the main text and in the method section.
Points 32-33 and figures 8 and 9: authors should merge the two figures because information are complementary. In addition, they will rewrite the corresponding analysis to simplify the read.
Point 36: It is not possible to maintain something and to be significantly different. In the text authors indicate that the viability of the cell exposed to JNK inhibitor is maintained compared the control. In the graph, we observed a significant reduction. Authors should check their results and adapt the text.
Point 37: I agree with the observation. However, the conclusion is not supported through a good statistic analysis. In addition the repeated answer is not a good justification.
Figure 10 - panels B and D: the quantitative analysis presented in the panel D is not at all representative of the results presented in the panel B. For example, the relative protein expression in the NAC condition shoud be higher compared to the 8a condition. I think that a problem occurred during the analysis.
Line 317: anti-cleave caspased 3 should be replace by anti-cleaved caspase 3
Point 43: I would like to have more details about the protocol. How many cell are used? How the authors prepare the samples? Are they collect the supernatant or they only scrapped the cells in the petri dish?
Section 3.9: authors should also add details in this section. How do they do the quantitative analysis? How do they prepare the cell fraction to analyze the mitochondrial cytochrome c fraction… What is the hardware to analyze their luminescence?
